# A Novel and Noninvasive Risk Assessment Score and Its Child-to-Adult Trajectories to Screen Subclinical Renal Damage in Middle Age

**DOI:** 10.3390/bioengineering10020257

**Published:** 2023-02-15

**Authors:** Chen Chen, Guanzhi Liu, Chao Chu, Wenling Zheng, Qiong Ma, Yueyuan Liao, Yu Yan, Yue Sun, Dan Wang, Jianjun Mu

**Affiliations:** 1Department of Cardiovascular Medicine, First Affiliated Hospital of Xi’an Jiaotong University, Xi’an 710061, China; 2Key Laboratory of Molecular Cardiology of Shaanxi Province, Xi’an 710061, China; 3Department of Orthopedics, The First Affiliated Hospital, Zhejiang University School of Medicine, Hangzhou 310009, China

**Keywords:** subclinical renal damage, machine learning, risk assessment tool, group-based trajectory modeling, screening strategy

## Abstract

This study aimed to develop a noninvasive, economical and effective subclinical renal damage (SRD) risk assessment tool to identify high-risk asymptomatic people from a large-scale population and improve current clinical SRD screening strategies. Based on the Hanzhong Adolescent Hypertension Cohort, SRD-associated variables were identified and the SRD risk assessment score model was established and further validated with machine learning algorithms. Longitudinal follow-up data were used to identify child-to-adult SRD risk score trajectories and to investigate the relationship between different trajectory groups and the incidence of SRD in middle age. Systolic blood pressure, diastolic blood pressure and body mass index were identified as SRD-associated variables. Based on these three variables, an SRD risk assessment score was developed, with excellent classification ability (AUC value of ROC curve: 0.778 for SRD estimation, 0.729 for 4-year SRD risk prediction), calibration (Hosmer—Lemeshow goodness-of-fit test *p* = 0.62 for SRD estimation, *p* = 0.34 for 4-year SRD risk prediction) and more potential clinical benefits. In addition, three child-to-adult SRD risk assessment score trajectories were identified: increasing, increasing-stable and stable. Further difference analysis and logistic regression analysis showed that these SRD risk assessment score trajectories were highly associated with the incidence of SRD in middle age. In brief, we constructed a novel and noninvasive SRD risk assessment tool with excellent performance to help identify high-risk asymptomatic people from a large-scale population and assist in SRD screening.

## 1. Introduction

Chronic kidney disease (CKD) is defined as abnormalities in kidney structure or function for at least 3 months with implications for health [1]. CKD has become a major public health concern due to its high prevalence and all-cause mortality [2,3]. The Global Burden of Disease Study reported that 697.5 million individuals suffered from CKD in 2017, with an overall prevalence of 9.1% [4]. A systematic review on the regional prevalence of CKD in Asia showed a substantial variation in CKD prevalence ranging from 7.0% in South Korea to 34.3% in Singapore, while China and India had the highest absolute number of people with CKD (159.8 million and 140.2 million, respectively) [5]. CKD is associated with a high risk of hospitalization, cardiovascular events, cognitive dysfunction, morbidity and all-cause mortality [6,7,8]. In addition, CKD may be accompanied by several other complications, including anemia, secondary hyperparathyroidism and electrolyte disturbances, creating substantial health care costs [9,10,11] and indicating the urgent need to prevent and manage renal damage progression at an early stage. Subclinical renal damage (SRD) is an early, asymptomatic renal abnormality characterized by a moderate increase in urinary albumin excretion or a moderate reduction in the glomerular filtration rate [12]. SRD can be defined by an estimated glomerular filtration rate (eGFR) between 30 and 60 mL/min/1.73 m^2^ or an elevated urinary albumin-to-creatinine ratio (uACR) more than 2.5 mg/mmol in men and 3.5 mg/mmol in women. The Hanzhong Adolescent Hypertension Cohort showed that the incidence of SRD in northern China was 13.1% [13]. Individuals with SRD tend to also have hypertension and diabetes mellitus [14], which further worsen renal function [1]. Early SRD detection and screening are essential to slow disease progression and reduce the risk of complications, morbidity and mortality, because the SRD condition can correspond to the stages of CKD (G3a stage, G3b stage in GFR Category and A2 stage, A3 stage in persistent albuminuria category) according to the 2012 KDIGO Clinical Practice Guideline for the Evaluation and Management of CKD [1]. Patients in these stages are mainly assessed as having moderately increased risk or high risk for concurrent complications and future outcomes; these are also the critical periods for early diagnosis and intervention for CKD. Currently, the detection and screening of renal function rely on biochemical assays with blood or urine samples. Serum creatinine can be used to evaluate eGFR and urine microalbumin, and creatinine can be used to evaluate uACR [15]. Biochemical analysis is the gold standard but is costly for long-term follow-up or large-scale population screening [16,17]. In addition, SRD is clinically asymptomatic and despite that renal function can be estimated by the measurement of serum creatinine concentration, urine protein or albumin concentration, it is still difficult to apply routine large-scale SRD screening, especially for asymptomatic adults, due to the lack of more economical and effective noninvasive risk assessment tools for SRD [1,18]. Hence, a simple and noninvasive risk assessment tool is urgently needed for SRD screening.

It has been reported that diabetes, hypertension, older age, obesity and smoking are independent risk factors for the development and progression of renal dysfunction [6,19,20,21]. Some studies have established prediction models for CKD risk based on these factors [22,23,24]. However, little attention has been given to the establishment of SRD risk assessment tools and the longitudinal observation of these tools. Recently, tracking trajectory patterns over time has accounted for dynamic changes and provided an important dimension for consideration. Group-based trajectory modeling is one of the approaches that considers variations in time [25]. Previous studies have suggested that long-term BP trajectories and long-term BMI trajectories are associated with the incidence of SRD [13,15,26]. However, single-variable trajectory practices are generally far from making full use of multivariate longitudinal data and the interrelationship of different variables.

In this study, we used data from Hanzhong Adolescent Hypertension Cohort to develop a noninvasive, economical and effective SRD risk assessment tool to identify high-risk asymptomatic people from a large-scale population and improve current clinical SRD screening strategies.

## 2. Materials and Methods

### 2.1. Cohorts and Participants

This study included participants from the Hanzhong Adolescent Hypertension Cohort, an ongoing prospective study initiated in 1987 that is focused on cardiovascular risk factor development. The Hanzhong Adolescent Hypertension Cohort recruited a total of 4623 schoolchildren from 26 rural sites of three towns in Hanzhong, Shaanxi, China in 1987, and several follow-ups were conducted in the following 30 years [27]. The inclusion criteria of the present study were as follows: aged 6–15 years in 1987, able to speak Mandarin to ensure effective communication, participated in the latest follow-up and had laboratory test data in 2017. For further trajectory analysis, complete blood pressure and BMI data during the 30-year follow-up were required. During the selection, individuals who had a history of myocardial infarction, heart failure, stroke, renal failure, or peripheral artery disease were excluded from the analysis. We conducted data collection in 1989, 1992, 1995, 2005, 2013 and 2017. In the 30 years of follow-up time, migration, death, mental illness and military service mainly contributed to the loss of follow-up. This study was clinically registered (NCT02734472) and approved by the Ethics Committee of First Affiliated Hospital of Xi’an Jiaotong University (Ethical Approval number: XJTU1AF2015LSL-047). All subjects gave written informed consent in advance. In addition, we obtained the consent of a parent/guardian for participants <18 years of age.

### 2.2. Anthropometric Measurements

Baseline clinical information, including demographic characteristics, histories of hypertension, hyperlipidemia, stroke and diabetes, history of cigarette smoking and alcohol consumption and cardiovascular complications, was collected using a standardized self-questionnaire. Body weight, height, waist circumference and hip circumference were measured by trained staff via standardized procedures. Body mass index (BMI) was calculated as weight in kilograms divided by height in meters squared (kilograms per meter squared). The average values of replicate measurements were used for further analysis.

### 2.3. Blood Pressure Measurements

Systolic and diastolic blood pressure were measured three times by trained and certified staff via WHO recommended procedures (in a seated position in a quiet and comfortable environment, 5-min rest before measurement, 2-min interval between examinations). Mean values of blood pressure were used for further analysis.

### 2.4. Biochemical Parameter Measurements

In this study, biochemical parameters, including total cholesterol (TC), triglyceride (TG), LDL cholesterol (LDL-C), HDL cholesterol (HDL-C), total bilirubin, serum creatinine, urinary uric acid (UA), creatinine and albumin levels, were measured according to standardized procedures. uACR (milligrams per millimole) was evaluated as urine albumin (in milligrams) divided by urine creatinine (in millimoles). eGFR was estimated by the Modification of Diet in Renal Disease (MDRD) calculation formula for Chinese patients with chronic kidney disease: eGFR = 175 × serum creatinine (in milligrams per deciliter) ^−1.234^ × age (in years) ^−0.179^ (×0.79 for females) [28].

### 2.5. Definitions

In this study, subclinical renal damage was defined as an eGFR between 30 and 60 mL/min/1.73 m^2^ or a uACR more than 2.5 mg/mmol in men and 3.5 mg/mmol in women [15]. Cigarette smokers were defined as subjects with >six months of smoking history during their lifetime (continuous or cumulative) [29]. Participants who reported that they drank alcohol (liquor, beer or wine) every day and that their alcohol consumption lasted for more than 6 months were defined as drinkers [30].

### 2.6. Statistical Analysis

To identify effective and reliable clinical parameters with high screening or early diagnostic value for SRD, we analyzed the cross-sectional data in 2017 (n = 2303) and provided a novel feature selection strategy by combining three machine learning methods (complete-case analyses), including LASSO regression, random forest and the SVM-REF algorithm. LASSO regression was performed via the R package “glmnet” [31], the random forest method was carried out by the R package “randomForest” and the SVM-REF approach was achieved by the R packages “sigFeature” and “e1071”. A logistic regression model was constructed based on the R package “rms”. The 2303 participants were randomly assigned to the training set (70%, n = 1611) and the internal validation set (30%, n = 692). The R package “pROC” was used to calculate the area under the curve (AUC) value of the receiver operating characteristic (ROC) curve [32]. In addition, calibration curve analysis and the Hosmer—Lemeshow goodness-of-fit test were performed using the R packages “rms” and “ResourceSelection”. Decision curve analysis was conducted by the R package “rmda” to evaluate the potential clinical application value and net benefit.

Next, group-based trajectory modeling was achieved by the “traj” package [33] in R software to identify the optimal number of subgroups with similar SRD risk score trajectories among those with complete blood pressure and BMI data during the 30-year follow-up in this cohort (n = 1048, complete-case analysis). Categorical data are summarized as frequencies and percentages. Continuous variables are reported as the mean ± standard deviation (if normally distributed) or the median (25th and 75th percentile ranges). Independent sample *t*-tests, one-way ANOVA, Mann—Whitney U tests and Kruskal—Wallis tests were performed for the difference analysis of continuous variables according to their group, distribution and variance. Logistic regression analysis was carried out by SPSS software (SPSS Inc., Chicago, IL, USA). Statistical significance was considered at a two-sided *p* value < 0.05 for all analyses.

## 3. Results

### 3.1. Study Population

The flow chart of the present study was shown in Figure 1. Overall, the latest follow-up data (the 7th follow-up, in 2017) of 2303 participants were included in the cross-sectional analysis to perform the machine learning feature selection and identify variables highly associated with SRD. Then, these 2303 participants were randomly assigned to the training set (70%, n = 1611) and the internal validation set (30%, n = 692). The training set was used to construct the SRD risk score model and the validation set was used to evaluate the SRD estimation performance. The data in 2013 (the 6th follow-up) were also included to evaluate the 4-year SRD risk prediction performance. The characteristics included in the model construction and validation of the participants in the training and internal validation sets are shown in Table 1. All variables have no significant differences between the training and internal validation sets, which suggested the data consistency and reasonableness of grouping. In addition, participants with complete blood pressure and BMI data during the 30-year follow-up were included in further group-based trajectory modeling analysis to identify the SRD risk score trajectories (n = 1048).

### 3.2. Feature Selection

A heatmap (Appendix A) showed the correlation among SRD and other 25 SRD-associated variables (anthropometric parameters, blood pressure level, biochemical parameters, diabetes history, etc.). Considering the data multicollinearity, it is necessary to conduct feature selection to identify the most important variables and then construct SRD risk models. In this study, we combined three machine learning algorithms to achieve accurate feature selection, including LASSO regression analysis, the random forest algorithm and the SVM-RFE algorithm. In LASSO regression analysis, 10-fold cross-validation was performed to detect the optimal AUC value and minimal parameters. Finally, we selected six features among 25 variables: systolic blood pressure, diastolic blood pressure, BMI, triglyceride, heart rate and diabetes (Figure 2A). The SVM-RFE algorithm was also used to achieve feature selection according to the optimal classification accuracy. Four variables were identified as key features: diastolic blood pressure, systolic blood pressure, BMI and body weight (Figure 2B). In addition, the random forest algorithm suggested six features (diastolic blood pressure, systolic blood pressure, BMI, triglyceride, serum chloride and serum potassium) to reach the minimum cross-validation error (Figure 2C). Meanwhile, based on the mean decrease in the Gini coefficient, the importance of variables in the random forest model were calculated (Figure 2D). Finally, by combining these three machine learning feature selection algorithms, we selected diastolic blood pressure, systolic blood pressure and BMI as hub variables for further analysis and model construction.

### 3.3. Construction and Validation of the SRD Risk Assessment Model

Logistic regression analysis was performed to establish an SRD risk assessment model based on data from the training set: SRD index = 0.020143 × SBP + 0.039718 × DBP + 0.063076 × BMI − 9.211994, SRD risk score = 1/(1 + e^−SRD index^). Meanwhile, a corresponding nomogram was constructed to achieve more efficient clinical application (Figure 3A). In detail, according to SBP, DBP and BMI data, total points can be calculated to evaluate the diagnostic possibility of SRD. High possibility indicates the need for further blood or urine testing to determine renal function, while low possibility indicates little need to take further tests, so as to achieve large-scale screening or self-monitoring. Next, we validated the classification ability of the model, and the AUC value of the ROC curve reached 0.778 (for SRD real-time estimation) and 0.729 (for 4-year SRD risk prediction) in the internal validation set (Figure 3B,C). The optimal cutoff value for SRD real-time estimation is 0.153, which leads to a sensitivity of 0.685 and specificity of 0.779. Meanwhile, the optimal cutoff value for 4-year SRD risk prediction is 0.117 which leads to a sensitivity of 0.767 and a specificity of 0.598. The calibration curve analysis and the Hosmer—Lemeshow goodness-of-fit test (*p* = 0.62 for SRD real-time estimation, *p* = 0.34 for SRD 4-year risk prediction) indicated that this model had good calibration in both SRD real-time estimation and SRD 4-year risk prediction (Figure 3D,E). In addition, as the SRD estimation decision curve analysis (DCA) showed, compared to the SRD screening decision strategies currently used in clinical practice, which mainly focus on the specific higher-risk conditions, such as hypertension, obesity and diabetes, more potential net benefit can be obtained in all ranges of risk thresholds using this SRD assessment model to assist in SRD screening decision making (Figure 3F,G). The results of the SRD 4-year risk prediction DCA also supported this conclusion. In fact, SBP, DBP and BMI data are easy to collect in clinical practice by noninvasive examination, which indicates that it is possible for our models to evaluate or predict the SRD risk and identify high-risk asymptomatic people from a large-scale population, which can improve existing SRD screening strategies.

### 3.4. SRD Risk Score Trajectory

SRD risk scores during the 30-year follow-up were calculated based on the diastolic blood pressure, systolic blood pressure and BMI data. Then, we performed group-based trajectory modeling analysis and identified three SRD risk score trajectory groups: stable, increasing-stable and increasing (Figure 4). The SRD risk scores of all three groups have trends of increasing with age from childhood to middle age and have similar slope increases before about 25 years old. After this age, the stable group (n = 376; 35.9%) endured relatively lower SRD risk score levels and SRD risk scores compared to the other two group, which continued to increase. The increasing-stable group (n = 404; 38.5%) was characterized by SRD risk scores increasing to a relatively higher level and then holding steady after about 40 years old. Meanwhile, the increasing group (n = 268; 25.6%) was characterized by a sustained increase from childhood to middle age and reached a higher level than both the stable group and increasing-stable group.

### 3.5. Cardiovascular Risk Factors for SRD Risk Score Trajectory Groups

Table 2 shows the data of partial anthropometry and biochemical indicator tests in 1987 and 2017 according to these three SRD risk score groups. Among these 1048 participants, 583 (55.6%) were males and 465 (44.4%) were females. The median age in 2017 was 43 years old. Differences in the proportion of males, age, incidence of hyperlipidemia, incidence of hypertension, current smoking, alcohol consumption, waist circumference, hip circumference, TC, TG, LDL-C, HDL-C, serum uric acid, serum creatinine, urine albumin and uACR were statistically significant (*p* <0.05). Occupation, education, marital status, incidence of carotid atherosclerosis, heart rate (both in 1987 and in 2007), urine uric acid (uUA) and eGFR were not significantly different. Individuals in the SRD risk score stable group were more likely to be females, and more likely to have a lower waist circumference, hip circumference, TC, TG, LDL-C and serum UA. In addition, the SRD risk score increasing group a higher incidence of hyperlipidemia and hypertension, as well as higher rate of current smoking and alcohol consumption.

### 3.6. Association between Novel SRD Risk Score Trajectories and Subclinical Renal Damage

SRD incidence was significantly different among the three SRD risk score groups (*p* < 0.05). Figure 5A shows that the SRD risk score increasing group had a higher SRD incidence rate in middle age (19%) compared to stable group (8.8%) and stable-increasing group (13.4%). We found that the uACR was significantly different among the three SRD risk score groups (*p* < 0.05), whereas the GFR was not significantly different (*p* = 0.26). The increasing group had a significantly higher uACR level (1.25 (0.74–2.34)) than the increasing-stable group (0.99 (0.64–1.96)) and the stable group (0.85 (0.57–1.33)). Additionally, the uACR levels between the stable, stable-increasing and increasing group were also significantly different (*p* = 0.002 for stable group compared to stable-increasing group, *p* < 0.001 for stable group compared to increasing group, *p* = 0.011 for stable-increasing group compared to increasing group). Moreover, the increasing group had a lower eGFR (94.3 (85.9–106.0)) compared to stable group (97.2 (86.2–107.0)) and stable-increasing group (97.7 (87.1–106.3)). The scatter diagrams of uACR levels and eGFR levels among these three groups are shown in Figure 5B,C. Next, logistic regression was performed to investigate the association between the SRD risk score trajectory groups and SRD incidence. The trajectory groups were defined as dummy independent variables, and the stable group was the control group in the logistic regression. Our results showed that the increasing group and increasing-stable group had significantly greater odds of SRD incidence in middle age than the stable group. The increasing-stable group had an OR of 1.6 (95% CI, 1.01 to 2.54), and the increasing group had an OR of 2.44 (95% CI, 1.53 to 3.91). The adjusted logistic regression model showed that ORs were slightly attenuated after adjustment for gender and age. The increasing-stable group had an OR of 1.53 (95% CI, 0.96 to 2.43), and the increasing group had an OR of 2.39 (95% CI, 1.49 to 3.84). Additional adjustment for waist circumference, hip circumference, TC, TG, LDL-C and HDL-C also attenuated the ORs. The increasing-stable group had an OR of 1.25 (95% CI, 0.77 to 2.05), and the increasing group had an OR of 1.75 (95% CI, 1.05 to 2.91). Finally, after further adjusting for the incidence of current smoking and alcohol consumption, the ORs of the increasing-stable group were 1.24 (95% CI, 0.76 to 2.03) and the ORs of the increasing group were 1.73 (95% CI, 1.04 to 2.89). These results indicated that these SRD risk score trajectories can serve as a strong predictor for the SRD incidence risk in middle age (Table 3). In addition, through long-term trajectory analysis, we can also demonstrate the good performance and reliability of this SRD risk assessment score in longitudinal observation.

## 4. Discussion

### 4.1. Main Findings

Three predictive factors (SBP, DBP and BMI) for SRD in middle age were identified using an integrated feature selection strategy. Based on these three predictive factors, a novel noninvasive SRD risk assessment model was established that showed excellent classification ability, calibration and potential clinical benefits for SRD estimation and SRD 4-year risk prediction. These results indicated that it is possible for our models to identify high-risk asymptomatic people from a large-scale population and help the clinical SRD early screening decision in middle age. Additionally, through subsequent cohort analysis, we identified three trajectory groups for this novel SRD risk assessment score using 30-year follow-up data. We found that the incidence of SRD in middle age and uACR levels were highly associated with these risk score trajectories. Further logistic regression analysis indicated that these SRD risk score trajectories can serve as a strong predictor for the SRD incidence risk in middle age. Therefore, longitudinal observation further confirmed the value of this risk score to generate individualized risk estimates and further participate in clinical screening decisions for SRD in middle age. In summary, we constructed a novel, simple and low-cost risk assessment tool for SRD screening, which presented good performance in predicting SRD risk in middle age. The convenience of this model makes it possible to assess the SRD risk of asymptomatic people and then carry out further SRD screening.

### 4.2. Prior Studies and the Focus of our Investigation

The detection and screening for SRD is critical because it can correspond to the CKD stages (G3a stage, G3b stage in GFR Category and A2 stage, A3 stage in persistent albuminuria category) which are associated with moderately increased risk (yellow risk) or high risk (orange risk) for the concurrent complications and future outcomes; these are also are the most critical periods for early diagnosis and intervention for CKD. However, SRD is usually asymptomatic until an advanced disease stage, and estimation methods of renal function, such as the measurement of serum creatinine concentration, urine protein or albumin concentration are costly for long-term follow-up or large-scale screening [34,35]. In current clinical practice, only patients with specific higher-risk conditions, such as hypertension, obesity and diabetes are recommended to be screened for renal function conditions or SRD. It is still difficult to apply routine SRD screening in a large-scale general population, especially for asymptomatic adults, due to the lack of a more economical and effective noninvasive risk assessment tool for SRD [1,36]. Therefore, a simple and noninvasive SRD risk assessment tool is urgently needed to assist in the SRD screening decision and improve large-scale SRD screening strategies. SRD is attributed to several risk factors, such as hypertension, diabetes, older age and obesity [37,38,39]. There have been numerous efforts to construct prediction models for the risk of decreasing eGFR in CKD [22,24]. However, the estimation or prediction of SRD can be more useful than only predicting a decrease in eGFR from the perspective of identifying the prognostic risk of CKD. In addition, too many variables and biochemical examination results were included in existing models, which complicated their translation to clinical practice for large-scale screening. Hence, in this study, we provided a novel feature-selection strategy by combining three machine learning methods, and first established an SRD risk assessment model calculated only by SBP, DBP and BMI data, which may have greater utility in clinical application. Additionally, our risk assessment model had better performance than those in previous studies: excellent classification ability (AUC value of the ROC curve: 0.778 for SRD estimation, 0.729 for 4-year SRD risk prediction in the validation set), calibration (Hosmer—Lemeshow goodness-of-fit test *p* = 0.62 for SRD estimation, *p* = 0.34 for 4-year SRD risk prediction) and potential clinical benefits.

In addition, most existing prediction models lack a longitudinal cohort analysis, such as group-based trajectory modeling analysis, which could reflect the relationship between model trajectory and SRD incidence [40,41]. Therefore, in the current study, we combined SBP, DBP and BMI data to calculate a novel SRD risk assessment score and then performed a trajectory analysis. Ultimately, three trajectory groups (increasing, increasing-stable, and stable) were identified based on 30-year follow-up data, and the incidence of SRD in middle age and uACR levels were highly associated with these risk score trajectories. Compared with the stable group, the increasing group and increasing-stable group had a significantly higher uACR. In addition, the results of the logistic regression showed that these three SRD risk assessment score trajectories could serve as ideal predictors of the incidence of SRD in middle age. Several other studies and some of our previous works have tried to investigate the relationship between SRD incidence and its risk-factor trajectories, such as SBP trajectory, DBP trajectory, MAP trajectory and BMI trajectory [13,15]. However, single-variable trajectory analyses have limitations because they ignore the interaction among multiple factors [42]. Hence, the group-based trajectory analysis for the SRD risk assessment score in the current work, which gives full consideration to the characteristics of SBP, DBP and BMI, is also a breakthrough for SRD-associated trajectory modeling analysis strategies.

### 4.3. Limitations and Future Directions

The present study used a community-based cohort followed for 30 years, which represents a large population. It is prospective in nature and consists of representative data from the general population. However, it should be noted that this study has the following limitations. First, our study used a racially-homogenous cohort from multiple rural areas in northern China, which limited the generalizability of our results, and validation using other cohorts with different backgrounds of ethnicities and populations will be performed in our further studies. Second, this work was not externally validated, which may also have limited the generalizability of our results. Notwithstanding this limitation, our study provided a novel SRD risk assessment tool that has both good performance in cross-sectional analysis and longitudinal analysis as well as the convenience of clinical application. In addition, to our knowledge, this is the first study to perform a group-based trajectory modeling longitudinal analysis for an SRD risk assessment tool, which revealed that realistic SRD outcomes in middle age correspond to the development trend of the risk score suggested by the SRD risk assessment model.

## 5. Conclusions

In conclusion, we used a large community-based cohort followed for 30 years to establish a novel, simple and low-cost SRD risk assessment tool and performed longitudinal group-based trajectory analysis for this tool. Internal validation suggested that our risk assessment model has excellent classification ability (AUC value of the ROC curve: 0.778 for SRD estimation, 0.729 for 4-year SRD risk prediction), calibration (Hosmer—Lemeshow goodness-of-fit test *p* = 0.62 for SRD estimation, *p* = 0.34 for 4-year SRD risk prediction) and potential clinical benefits. Further longitudinal trajectory analysis also confirmed the reliability of this SRD risk assessment score. Considering the good clinical utility, simplicity and convenience as well as the excellent performance of our model, it can identify high-risk asymptomatic people from a large-scale population and improve current clinical SRD screening strategies.

## Figures and Tables

**Figure 1 bioengineering-10-00257-f001:**
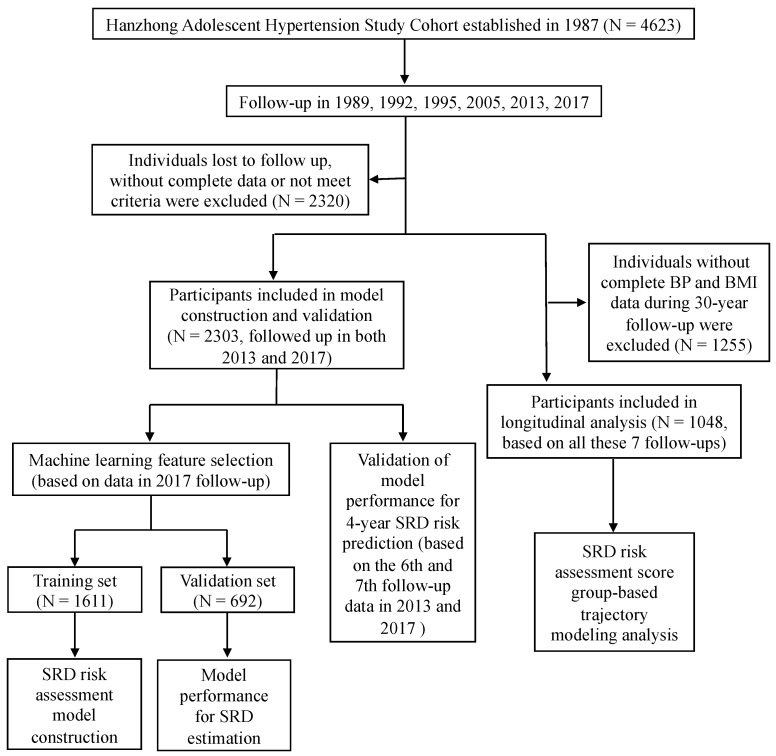
Flow chart of research design.

**Figure 2 bioengineering-10-00257-f002:**
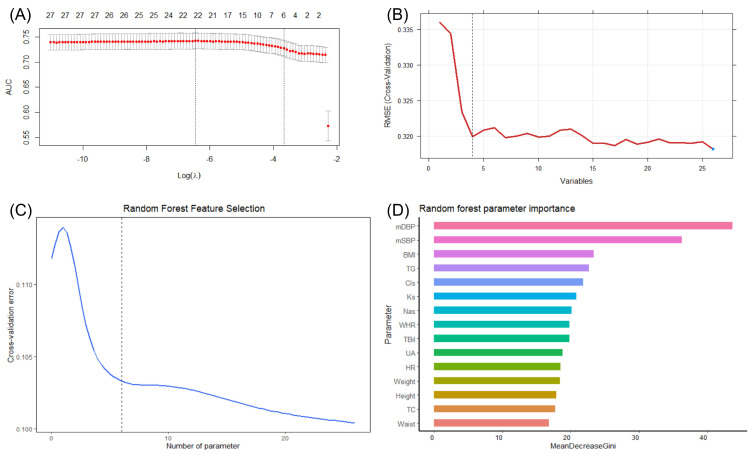
Machine learning feature selection strategies in this study. (**A**) LASSO regression analysis. Six features were identified including SBP, DBP, BMI, triglyceride, heart rate and diabetes. (**B**) SVM-RFE algorithm feature selection. Four features were identified: DBP, SBP, BMI and body weight. (**C**) Random forest algorithm feature selection. Six features were selected: DBP, SBP, BMI, triglyceride, serum chloride and serum potassium. (**D**) Importance of the parameters was assessed by a random forest algorithm. AUC, area under the curve; RMSE, root mean square error; mDBP, mean diastolic blood pressure; mSBP, mean systolic blood pressure; BMI, body mass index; TG, triglyceride; Cls, serum chloride iion; Ks, serum potassium; Nas, serum sodium; WHR, weight-to-height ratio; TBil, total bilirubin; UA, uric acid; HR, heart rate; TC, total cholesterol.

**Figure 3 bioengineering-10-00257-f003:**
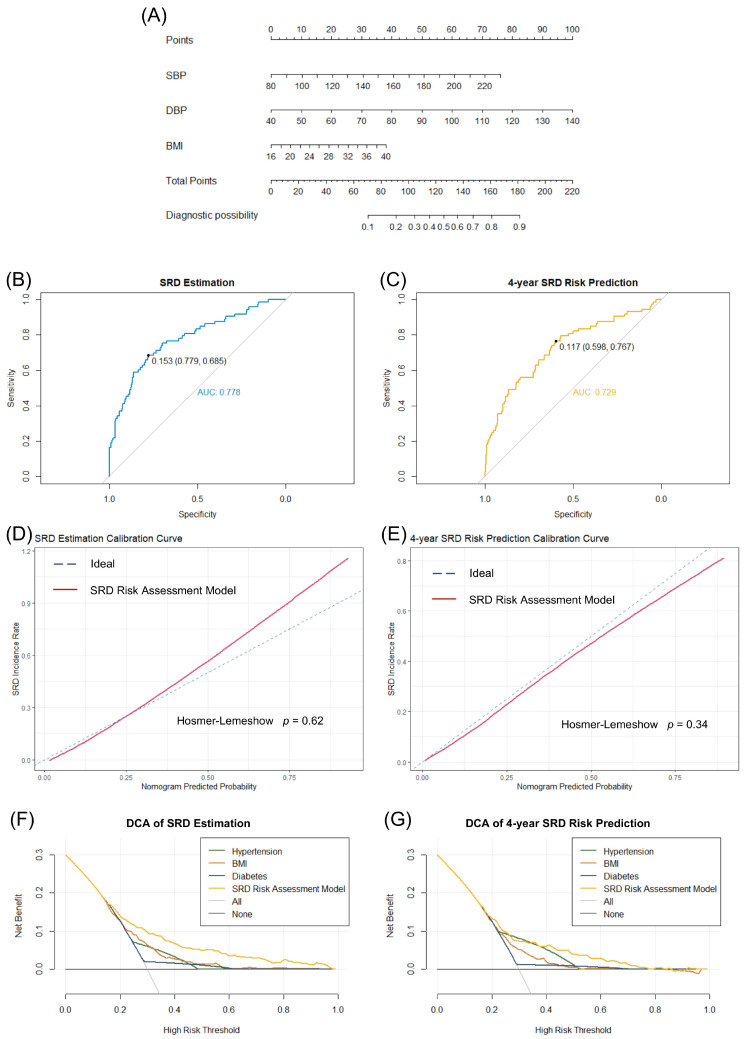
SRD risk assessment model construction and validation. (**A**) Nomogram for SRD risk. Diagnostic possibility can be calculated based on SBP, DBP and BMI. (**B**,**C**) AUC value of the ROC curve in the internal validation set. The SRD estimation AUC value can reach 0.778 and the 4-year SRD risk prediction AUC value can reach 0.729. (**D**,**E**) Calibration analysis for this SRD risk assessment model. (**F**,**G**) Decision curve analysis for hypertension, diabetes, BMI and this SRD risk assessment model, which showed this model had greater potential clinical benefits than each individual variable used to assess SRD risk in current clinical practice such as hypertension, diabetes and BMI. SRD, subclinical renal damage; SBP, systolic blood pressure; DBP, diastolic blood pressure; BMI, body mass index; AUC, area under the curve; DCA, decision curve analysis.

**Figure 4 bioengineering-10-00257-f004:**
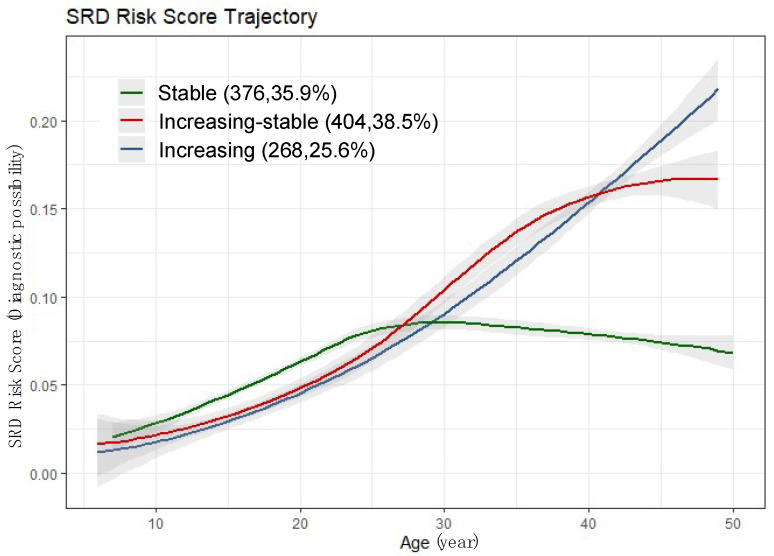
Three SRD risk score trajectory groups identified in this study using group-based trajectory modeling analysis: stable group, increasing-stable group and increasing group. SRD, subclinical renal damage.

**Figure 5 bioengineering-10-00257-f005:**
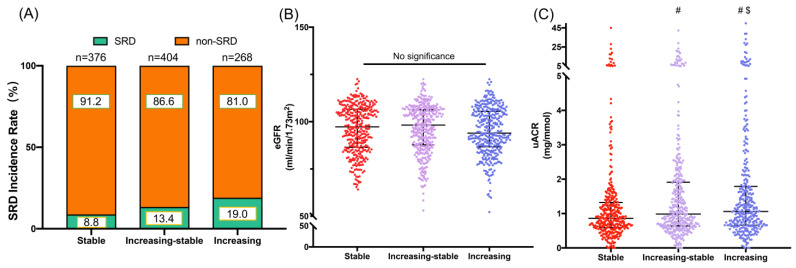
Renal damage of different trajectories groups. (**A**) SRD incidence rate among the three SRD risk score trajectory groups. (**B**,**C**) Scatter diagrams of eGFR levels and uACR levels among these three SRD risk score trajectory groups. SRD, subclinical renal damage; eGFR, estimated glomerular filtration rate; uACR, urinary albumin-to-creatinine ratio. ^#^ *p* < 0.05 vs. stable group and ^$^ *p* < 0.05 vs. increasing-stable group.

**Table 1 bioengineering-10-00257-t001:** Characteristics of the participants in the training and internal validation sets.

Characteristics	Total	Training Set	Internal Validation Set	*p* Value
SBP (mmHg)	121.3 (112.7–131.3)	121.7 (113.0–131.3)	120.8 (112.0–131.3)	0.363
DBP (mmHg)	76.0 (69.3–84.3)	76.3 (70.0–84.3)	75.3 (68.3–84.7)	0.096
BMI (kg/cm^2^)	23.8 (21.9–26.0)	23.8 (21.9–26.2)	23.8 (21.9–25.6)	0.397
eGFR (mL/min per 1.73 m^2^)	96.9 (87.1–106.1)	96.5 (86.8–105.8)	98.2 (88.0–106.6)	0.096
uACR (mg/mmol)	0.95 (0.62–1.68)	0.95 (0.62–1.69)	0.96 (0.63–1.65)	0.939
SRD (n, %)	276 (13.2)	203 (13.9)	73 (11.7)	0.177

SBP, systolic blood pressure; DBP, diastolic blood pressure; BMI, body mass index; eGFR, estimated glomerular filtration rate; uACR, urinary albumin-to-creatinine ratio; SRD, subclinical renal damage.

**Table 2 bioengineering-10-00257-t002:** Demographic characteristics and cardiovascular risk factors by the SRD risk score trajectory groups.

	Total	Stable	Increasing-Stable	Increasing	*p* Value
Male (%)	583	169 (45.1)	258 (63.7)	156 (58.2)	<0.001
Age (years)	43.0 (41.0–46.0)	43.0 (41.0–46.0)	43.0 (40.0–45.0)	43.0 (41.0–45.0)	0.049
Occupation (%)	1011				0.383
Farmer	408	146 (40.1)	157 (40.3)	105 (40.9)	
Worker	194	63 (17.3)	82 (21.0)	49 (19.1)	
Business	81	35 (9.6)	30 (7.7)	16 (6.2)	
Governor	21	5 (1.4)	13 (3.4)	3 (1.2)	
Other	307	115 (31.6)	108 (27.7)	84 (32.7)	
Marital status (%)	1041				0.064
Unmarried or other	15	4 (1.1)	8 (2.1)	3 (1.2)	
Married	1015	365 (97.1)	387 (97.0)	263 (98.9)	
Divorced	11	7 (1.9)	4 (1.0)	0 (0.0)	
Education (%)	1016				0.553
Primary school or less	73	24 (6.6)	27 (6.9)	22 (18.7)	
Middle school	628	221 (60.5)	240 (61.2)	167 (64.5)	
High school	226	82 (22.5)	92 (23.5)	52 (20.1)	
College or more	89	38 (10.4)	33 (8.4)	18 (6.9)	
Current smoking (%)	450	126 (34.8)	200 (51.9)	124 (49.2)	<0.001
Alcohol consumption (%)	321	96 (26.5)	141 (36.6)	84 (33.3)	0.011
SRD (%)	138	33 (8.8)	54 (13.4)	51 (19.0)	0.001
AS (%)	139	48 (12.9)	55 (13.9)	36 (13.7)	0.922
Hyperlipidemia	424	119 (31.6)	170 (42.1)	135 (50.4)	<0.001
Hypertension	172	10 (2.7)	65 (16.1)	97 (36.2)	<0.001
Heart rate 1987 (beats/min)	78.0 (72.0–84.0)	78.0 (72.0–84.0)	78.0 (72.0–84.0)	78.0 (72.0–84.0)	0.983
Heart rate 2017 (beats/min)	73.0 (66.0–80.0)	72.5 (66.0–79.0)	73.0 (66.0–80.0)	75.0 (69.0–82.0)	0.072
Waist (cm)	84.8 (78.2–92.2)	80.8 (75.5–87.2)	87.0 (79.7–94.3)	89.4 (82.4–95.5)	<0.001
Hips (cm)	92.2 (88.8–95.9)	90.7 (87.7–93.4)	93.4 (89.5–97.0)	93.7 (90.4–97.0)	<0.001
TC (mmol/L)	4.48 (4.03–5.00)	4.40 (3.92–4.87)	4.49 (4.02–5.08)	4.58 (4.17–5.18)	0.001
TG (mmol/L)	1.39 (1.01–2.01)	1.20 (0.89–1.66)	1.44 (1.08–2.03)	1.64 (1.13–2.44)	<0.001
LDL–C (mmol/L)	2.49 (2.11–2.88)	2.44 (2.05–2.78)	2.48 (2.13–2.95)	2.55 (2.22–3.00)	0.006
HDL-C (mmol/L)	1.13 (0.99–1.33)	1.20 (1.02–1.42)	1.12 (0.98–1.29)	1.09 (0.95–1.29)	<0.001
Serum uric acid (μmol/L)	283.2 (226.2–338.8)	264.9 (212.5–316.8)	300.7 (239.7–352.6)	293.8 (243.3–352.2)	<0.001
Urine uric acid (μmol/L)	1298.5 (914.8–1984.5)	1291.5 (897.5–1994.5)	1317.0 (981.5–1951.0)	1283.0 (889.0–2090.0)	0.268
Serum creatinine (μmol/L)	76.3 (66.7–86.8)	73.7 (65.3–82.9)	78.8 (68.6–88.8)	77.0 (69.7–88.0)	<0.001
Urine albumin (mg/L)	8.0 (4.1–13.7)	6.4 (3.1–11.1)	9.0 (4.8–14.2)	9.2 (5.2–22.5)	<0.001
eGFR (mL/min per 1.73 m^2^)	97.2 (87.0–106.3)	97.2 (86.2–107.0)	97.7 (87.1–106.3)	94.3 (85.9–106.0)	0.260
uACR (mg/mmol)	0.98 (0.64–1.72)	0.85 (0.57–1.33)	0.99 (0.64–1.96)	1.25 (0.74–2.34)	<0.001

AS, atherosclerosis; TC, total cholesterol; TG, triglycerides; LDL-C, low density lipoprotein cholesterol; HDL-C, high density lipoprotein cholesterol.

**Table 3 bioengineering-10-00257-t003:** Adjusted odds ratios and 95% confidence intervals of the association of SRD risk score trajectory groups with subclinical kidney damage.

Trajectory Groups	No. of Subjects with SRD in 2017	Unadjusted	Model 1	Model 2	Model 3
Stable	33 (8.8)	1.00	1.00	1.00	1.00
Increasing-stable	54 (13.4)	1.60 (1.01–2.54)	1.53 (0.96–2.43)	1.25 (0.77–2.05)	1.24 (0.76–2.03)
Increasing	51 (19.0)	2.44 (1.53–3.91)	2.39 (1.49–3.84)	1.75 (1.05–2.91)	1.73 (1.04–2.89)

Model 1 = gender, age in 2017. Model 2 = Model 1 + waist circumference, hip circumference, TC, TG, LDL-C and HDL-C in 2017. Model 3 = Model 2 + current smoking and alcohol consumption in 2017.

## Data Availability

Not applicable.

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
