# Peer review of "A Novel and Noninvasive Risk Assessment Score and Its Child-to-Adult Trajectories to Screen Subclinical Renal Damage in Middle Age"

_bioengineering, 2023, doi:10.3390/bioengineering10020257_

Round 1

Reviewer 1 Report

The word from Chen Chen et al. intitled 

A Novel and Noninvasive Risk Assessment Score and Its Child-to-Adult Trajectories to Screen Subclinical Renal Dam- 3 age in Middle Age  

describes a novel, straightforward, and affordable SRD risk assessment instrument and did longitudinal 406 group-based trajectory analysis for this measure using a sizable community-based cohort followed over 30 years. This by itself is an extraordinary achievement and has enormous value. The authors found tha their risk assessment model has excellent calibration (Hosmer-Lemeshow goodness-of-fit test P = 0.62 for SRD estimation, P = 0.34 for four-year SRD risk prediction), classification ability (AUC value of the ROC curve: 0.778 408 for SRD estimation, 0.729 for four-year SRD risk prediction), and potential clinical benefits. The validity of this SRD risk assessment score was further supported by longitudinal trajectory analysis. Their model, had outstanding performance, good clinical value, and simplicity and ease, can detect high-risk asymptomatic individuals from huge populations and enhance existing clinical SRD screening procedures.

For that reason, I highly recommend its publication after the following minor issues will be corrected:

Page 9, line 268: Use male and females instead of boys and girls.

Figure 5, panel C: The simbols #$ is not in the figure caption, please fix it.

Page 14, line 432: Data Availability Statement is very important here due to extension of the study, I would suggest to share the data or make clear why this is not possible.

Reviewer 2 Report

This is an interesting article and I recommend this should be published. 

Reviewer 3 Report

Dear Authors,

An interesting manuscript, thank you. However, I have some minor objections:

1) Introduction. I would like to ask the authors to end this section with an aim and to move Line 86-91 to the end of Discussion, if necessary;

2) Material and methods.2.1. please, define the clear exclusion criteria also; - additionally, indicate why so important is to know the Mandarin (in the inclusion criteria), this should be shortly explained here;

2.2. add the used questionnaire as a Supplementary File, please, this might be interesting for the readers;

3) Results. Decipher, please, abbreviations used in Figs. 2, 3, 4.

Thank you for the references, - all are from this century whats quite rare in scientific manuscripts.
